# Care When It Counts: Establishing Trauma-Sensitive Care as a Preventative Approach for 0–3-Year-Old Children Suffering from Trauma and Chronic Stress

**DOI:** 10.3390/children10061035

**Published:** 2023-06-08

**Authors:** Serafine Dierickx, Laura Malisse, Elisa Bisagno, Alessia Cadamuro, Sarah Van Haeken, Dorien Wuyts, Zane Linde-Ozola, Annija Kandãte, Dorottya Morva, Monika Rozsa, Andrea Gruber, Johanna M. C. Blom, Laura Giovanna De Fazio, Dima Bou Mosleh, Dóra Varga-Sabján, Anne Groenen

**Affiliations:** 1Expertise Centre Resilient People, University Colleges Leuven-Limburg (UCLL), 3000 Leuven, Belgium; 2Faculty of Psychology and Educational Sciences, Katholieke Universiteit Leuven (KU Leuven), 3000 Leuven, Belgium; 3Department of Biomedical, Metabolic and Neural Sciences, University of Modena and Reggio Emilia, 41121 Modena, Italy; 4REALIFE Research Group, Department of Development and Regeneration, Faculty of Medicine, Women and Child KU Leuven, 3000 Leuven, Belgium; 5Department of Anthropology, Faculty of Humanities, University of Latvia, LV-1586 Riga, Latvia; 6Center Dardedze, LV-1002 Riga, Latvia; 7Pressley Ridge Hungary Foundation, 1142 Budapest, Hungary; 8Department of Law, University of Modena and Reggio Emilia, 41121 Modena, Italy; 9Centre for Neuroscience and Neurotechnology, University of Modena and Reggio Emilia, 41121 Modena, Italy; 10Leuven Institute of Criminology (LINC), Katholieke Universiteit Leuven (KU Leuven), 3000 Leuven, Belgium

**Keywords:** trauma-sensitive care, trauma, chronic stress, children, professional caregivers

## Abstract

Adverse childhood experiences are an important societal concern. Children aged 0–3 are particularly vulnerable to unpredictable chronic stress due to the critical period for brain development and attachment. Trauma-sensitive care is a preventative approach to reduce the burden of stressful experiences by committing to positive relationships. Professional caregivers are ideally placed to offer trauma-sensitive care; however, earlier research reveals that the tools they need to consciously apply trauma-sensitive care principles are missing. The current study organized living labs (co-creative research method) to present trauma-sensitive care as a preventative approach aimed at children aged 0–3. Two living labs were organized in Belgium and Hungary, where professional caregivers collaborated to create a protocol that offers guidelines on how to implement trauma-sensitive care. The resulting protocol included a theoretical foundation on trauma as well as a translation of these guidelines into practical recommendations. The protocol was evaluated by incorporating it into a training intervention delivered to 100 professional caregivers from childcare organizations across four European countries. The protocol received positive feedback from participants, with results indicating a self-reported increase in knowledge, attitude and practice of trauma-sensitive care principles. We conclude that this trauma-sensitive care protocol is a promising answer to the needs of professional caregivers working with children aged 0–3.

## 1. Introduction

Adverse childhood experiences (ACEs) are defined as the chronic and repeated experiences of abuse, neglect or other types of severe family dysfunction before the age of 18 [1]. Being confronted with ACEs at a young age can disrupt the child’s typical developmental processes, leading to both immediate and long-term adversities in terms of mental and physical health, such as an unhealthy lifestyle, various diseases and chronic conditions [1,2,3,4]. Trauma is also a possible result of adverse experiences [5]. Understanding to what extent an adverse event will impact a person’s health and life course trajectory is not straightforward. Moderating factors such as severity, duration and nature of the event [6] influence the outcome. Interestingly, the way stress is manifested is also greatly determined by timing and the developmental phase that the child is going through at the time of the ACE [6]. 

Young children are more vulnerable to the development of trauma symptoms (e.g., hypervigilance, emotional unavailability) due to rapid growth and changes occurring in the brain during the first five years of their lives [6,7]. During early childhood the foundations are laid for the ensuing stages of cognitive, emotional and social development [6]. An extreme, prolonged activation of the body’s stress response system due to exposure to ACEs can affect critical regions in the brain responsible for emotion regulation, executive function and memory, possibly leading to trauma [2,3,4]. As brain development is cumulative, building on earlier acquired knowledge and skills to develop more complex abilities, getting a rough start in life can therefore lead to enduring disparities in learning behavior and health [8], making it difficult for the individual to reach their full potential as a healthy and productive member of society [2,8]. Consequently, preventative and intervening measures aimed at addressing adverse experiences and their effects during early childhood (i.e., the first thousand days) could have the biggest possible return for the individual, their families and society at large [8]. 

Though accurate prevalence numbers are difficult to obtain due to the hidden nature of child maltreatment, research indicates the highest rates of child maltreatment occur in children aged 0–3 [9]. Child maltreatment in the youngest children has a particularly high risk of remaining undetected because these children are unable to communicate their experiences and thus they are more dependent on their primary caregivers (mostly parents) [10,11]. 

For young children, interactions with significant and caring adults present in the child’s life draw the blueprints of the way they view themselves, others and the world [12,13] and trauma can intrude on these very imprints [14]. Infants and toddlers need supportive adults to help them regulate states of emotional arousal and stress [15]. According to McConnico et al. [14], attuned interaction with a caregiver can serve as a protective factor by “facilitating neural activity in the brain that increases the likelihood of adaptive development of the stress response system” (pp. 39). Over time, children learn to self-regulate [15]. However, overwhelming states of stress can lead to long-term patterns of dysregulation in the stress response system [8]. Children in a chronic dysregulated state can be more difficult to soothe and will often be labelled as “behaving badly” by caregivers, leading to further disconnection and mutually dysregulating interactional cycles with parents and caregivers [8]. 

During the first three years of life, many children in Europe encounter professional caregivers working at childcare organizations such as day-care, preschool, nursery, or kindergarten, who routinely work with a small group of the same children during the day. These professional caregivers will undoubtedly be confronted with children suffering the consequences of chronic stress considering its high prevalence rate [9].. In Belgium and Hungary, the countries where the trauma-sensitive care protocol was created, childcare organizations are not formally part of healthcare. Consequently, these organizations are often excluded from existing initiatives towards the prevention and treatment of child maltreatment. Current initiatives focus mostly on specialized healthcare organizations, making professional caregivers an untapped resource in combating trauma. 

Professional caregivers are in a unique position to recognize, treat and prevent child maltreatment, i.e., provide trauma-sensitive care [16]. Though there are many definitions, trauma-sensitive care is generally characterized as a preventative approach to reduce the burden of chronic stress by committing to positive relations [17]. Unfortunately, professional caregivers in Europe indicate they currently lack the knowledge and skills to integrate trauma-sensitive care into their daily practice [16]. The present study builds on data based on findings from phase 1 and phase 2 of a research project (see Figure 1) which have been documented by [16]. In those two phases, desk research and focus group discussions in four European partner countries (Belgium, Hungary, Latvia and Italy) demonstrated that although professional caregivers from childcare organizations have developed measures to cope with trauma symptoms, these are often based on gut feelings and experience, not on formal guidelines (see [16]., for a detailed description of the current use of trauma-sensitive care in the partner countries involved). There is a lack of guidelines or trainings available on screening, referral and responding to signs of maltreatment in young children [16,18]. Concurrently, there are no formal specific training interventions of trauma-sensitive care attuned to professional caregivers working with children between the ages of 0 and 3 [16]. 

Huang et al. [19] presume that a trauma-sensitive care approach should include: (1) a concise understanding of the widespread impact of trauma and potential paths to recovery; (2) knowledge of the signs and symptoms of trauma from a systemic perspective; and (3) an embeddedness of a perspective on health and wellbeing in the policy and general structure of an organization that integrates warmth, support, safety and predictability in order to prevent and address trauma. The routine and structure of professional caregiver organizations make them particularly well-placed to provide a safe and supportive environment for stimulating healthy development for all infants and for encouraging post-traumatic healing for those most vulnerable [19]. The sustained physiological regulation that stems from engaging in trauma-sensitive care, in combination with the positive relations and routine usually inherent in childcare organizations can directly and positively impact the recovery from trauma and the development of the young child [19,20,21]. This highlights the necessity of providing childcare organizations with support in meeting the needs of children and families who have been exposed to trauma and chronic stress [22,23]. 

### The Present Study

At present, there is a lack of information available concerning trauma-sensitive care for children aged 0–3 years old. Resources designed to support professional caregivers working with young children in applying a trauma-sensitive care (TSC) model are even more rare [16]. The present study aims at enhancing the knowledge, skills and attitudes of professional caregivers working with children aged 0–3 and their families, using trauma-sensitive care as a way of meeting the needs of those who may have experienced trauma or chronic stress. 

This study describes the co-creation of a trauma-sensitive care protocol specifically designed for and by professional caregivers working with children aged 0–3 years old. The specific purpose was to promote a culture shift in the childcare sector by building capacity of professional caregivers to prevent and address child maltreatment in a trauma-sensitive manner, thereby creating expertise and knowledge. However, the scope of trauma-sensitive care extends beyond child maltreatment and neglect to other situations in which young children may experience chronic stress. We examine the initial implementation phase of this co-creational process that was shaped as a living lab centered on professional caregivers and their stakeholders. A variety of data sources and methods were used to track the developmental process of the trauma-sensitive care protocol. The data were used to guide understanding and decision making during the co-creational and implementation phases.

## 2. Materials and Methods

### 2.1. Participants and Procedure

The present study is part of a larger ECLIPS project (Figure 1).ECLIPS is an acronym for ‘Enhancing the Capacity to combat child abuse through an Integral training and Protocol for childcare professionalS’. The. The project aimed at supporting professional caregivers working with children aged 0–3, especially childcare professionals, in detecting child maltreatment, in referring effectively and efficiently, and in coping with trauma due to child maltreatment amongst children, parents and other professional caregivers. The present study focuses on the last two phases of the project (see Figure 1), phase 3 and phase 4. Phase 1 and phase 2 have been described by Dierickx et al. [16]. In phase 3, a sample of professionals and stakeholders was involved to co-create a trauma-sensitive care protocol by means of living labs. Subsequently, in phase 4, training was offered to 100 childcare professionals from four countries (Hungary, Belgium, Latvia and Italy) in order to support the application of the trauma-sensitive care protocol. The participants of the training intervention were invited to evaluate the trauma-sensitive care protocol based on usability and eligibility. 

#### 2.1.1. Living Lab Method

The study adopted a living lab method. A living lab is defined as a practice-driven organization that facilitates and fosters open, collaborative innovation as well as a real-life environment or arena where both open innovation and user-innovation processes can be studied and where innovative solutions are developed [23]. A living lab is a co-creative research method in which we develop, test and use a product or service together with the end-users. The potential for social and innovative development through co-creation in all disciplines of society is widely recognized at the local and international level [24]. 

The aim of the living lab sessions of this project was to develop a trauma-sensitive care protocol for professional caregivers. Participants were 10–15 professional caregivers (end-users, i.e., childcare professionals but also pediatricians and health visitors), parents and policy members from Belgium (Living lab 1) and Hungary (Living lab 2). The other two project partners from Italy and Latvia organized two living labs independently from Belgium and Hungary because they focused on another topic, namely detecting and referring child maltreatment and trauma [25]. A convenience sample of 10–15 participants was composed within the region of the researcher’s workplace. They participated in eight sessions that took place between May 2021 and February 2022. Invitations to participate in the study were sent to childcare organizations to distribute and discuss among their professional caregivers. After agreeing with the informed consent, participants were invited to take part in one session each month (with an exception for the months of July and August). 

The first step (sessions 1 and 2) of the living lab, i.e., exploration, comprised an exploration of needs and opportunities for trauma-sensitive care by professional caregivers and for children aged 0–3. The first step also included a content analysis of the collected data during Phase 1 and Phase 2 of the research (see Figure 1). This analysis led to the endorsement of themes and preconditions of the development of a trauma-sensitive care protocol. The second step was labelled “co-creation” (sessions 3–6) and involved the participants’ active input of what should and should not be included in the protocol. The content and structure of the protocol were determined. The co-creational step led to a prototype version of the trauma-sensitive care protocol. During the pilot step (sessions 7 and 8), the participants evaluated the prototype in their work environment and suggested alterations. The alterations were discussed and documented, followed by another content analysis of the collected data. Changes were made accordingly. The pilot step led to a finalized trauma-sensitive care protocol.

The living labs were organized by a practice-based institution in Hungary and a knowledge-based institution in Belgium (Flanders). Both living labs followed the same structure and planning. In between each session, the organizing partners discussed the results of the sessions to exchange feedback and finally agree on a shared protocol. After the living lab sessions, the trauma-sensitive protocol was reviewed by external experts (i.e., psychologists, social workers, policy members, and project partners from the knowledge-based institution and the practice-based institution). The researchers integrated the feedback offered on language use and content of the experts and finalized the trauma-sensitive care protocol. 

#### 2.1.2. Training Intervention

The finalized trauma-sensitive care protocol was embedded in a one-day training intervention. In this one-day training intervention, 100 adult professional caregivers working within the childcare sector in the four project partners’ countries (i.e., Hungary, Belgium, Latvia and Italy; see Table 1) participated. The professional caregivers were recruited via snowball sampling to participate in a training on screening for signals of child maltreatment, referring families to the appropriate care, and engaging in trauma-sensitive care for children aged 0–3, their families and the professional environment. After giving informed consent, participants were invited to take part in the study. In each training session, at least one member of the living lab sessions took part. 

Each project partner organized two trainings for groups of professional caregivers (*n* = 8–15 per group; see Table 1). A one-day training was offered by two project researchers with trainer experience in the native language. The day was comprised of three components, including firstly, theoretical knowledge about child maltreatment and trauma in the age group 0–3 years old. Secondly, the importance of screening, referral and trauma-sensitive care were explained by means of theory and exercises. Finally, the trauma-sensitive care protocol (developed in the living lab sessions) was elucidated in detail and use of the protocol was practiced. After the one-day training, the participants received a copy of the trauma-sensitive care protocol for personal use. 

### 2.2. Measures

The professional caregivers evaluated their own knowledge, skills and attitude through a self-report questionnaire (see Appendix A) at baseline (T0), one month after the training intervention (T1) and three months after the training intervention (T2). Only the scales relevant to this study are retained. The items reflected the level of agreement with relevant statements, to be rated on a 5-point Likert scale, (1 = strongly disagree and 5 = strongly agree). The subscale Knowledge was defined as the awareness of child maltreatment and its impact (7 items, e.g., “Trauma is not only related to physical and sexual abuse but also to psychological maltreatment and neglect”). The subscale Attitude covers the role of professionals towards children who are struggling from extreme stress or trauma (4 items, e.g., “Childcare professionals can play an important role in screening and referring child abuse”). Practice (consisting of Infant Care, Parent Interaction, Self-Care and Team environment) is defined as the set of skills an individual has to cope with signs of stress or child maltreatment concerning infant care, parent interaction, self-care and team environment (13 items, e.g., “I know strategies to engage and soothe children during moments of crisis”). Evaluation of the trauma-sensitive care protocol represents an assessment of the usability and eligibility of the developed protocol (3 items, e.g., “I found the trauma-sensitive care protocol easy to comprehend”). Finally, General feedback on the training includes the assessment of the overall training that was developed by the researchers, based on the protocol (4 items, e.g., “The content of the program aligned with my expectations”). The questionnaire was assessed anonymously. The questionnaire was translated into the native languages of the project partners to ensure that the participants could complete the assessment in a language they were familiar with. 

### 2.3. Analyses

The living labs included in the present study are part of the developmental process of the trauma-sensitive protocol. The analyses of the results describe the chapters that were created in the living lab sessions. The theoretical frameworks that were integrated and their added value to the protocol are included.

SPSS version 28 was used to analyze the data of the self-report questionnaire. To assess evolution over time, mean scores and standard deviations for the scales Knowledge, Skills and Attitude are included for all three Measuring moments (Mm). The mean scores and standard deviations of Evaluation of the TSC Protocol and General feedback on the training are included for T1 and T2, as these two scales were not assessed at T0 (baseline). Additionally, two-way ANOVA (measuring moment*country) was used to examine the self-reported differences in scores from T0 to T1 to T2 and for all four countries (Hungary, Belgium, Latvia, and Italy).

## 3. Results

### 3.1. Living Lab Method

The open-access protocol can be found on this website: https://eclipsproject.eu/wp-content/uploads/2022/04/eclips-trauma-informed-care-protocol-screen.pdf, accessed on 30 March 2023. The trauma-sensitive protocol is aimed at supporting professional caregivers in offering trauma-sensitive care to children aged 0–3 years old. Based on the results of Phase 1 and Phase 2 of the project (described in [16]), the trauma-sensitive care protocol should integrate theoretical knowledge on trauma and possible consequences, sensitization on the impact of professional care on young children struggling with stress and trauma, and practical guidelines on how to use trauma-sensitive care, both on the level of an organization and on the level of an individual. The finalized protocol consists of five chapters. After each chapter, reflective exercises are added to enhance the translation from theory to practice. To further facilitate mastery of the TSC principles, interferences are spread throughout each chapter in which professionals can read about what they can do in the context of the explained theory. These reflections and tips were added to increase the usability and practice-oriented focus of the trauma-sensitive protocol.

#### 3.1.1. Chapter 1: The Meaning of Adverse Childhood Experiences and Trauma

The first chapter focuses on increasing insights about stress, adverse childhood experiences and trauma in children aged 0–3 years old. The chapter opens with the definition of stress as a dimensional construct ranging from “productive stress”, that boosts individual growth, to “unproductive or toxic stress” that can have a detrimental effect on our wellbeing and health. The Window of Tolerance [26] then illustrates the dimensionality of our individual stress levels. Subsequently, Adverse Childhood Experiences or ACEs [1] are described as examples of how unproductive or toxic stress can have a detrimental effect on individuals. The chapter continues to formulate and clarify the construct of trauma and its complexity, as well as trauma within the child-caregiver relationship. Finally, Perry and Ablon’s [27] theoretical intervention of regulate, relate, reason or respectively, empathize, share, and collaborate, is presented.

#### 3.1.2. Chapter 2: The Impact of Trauma on Young Children

The second chapter of the trauma-sensitive care protocol focuses on how trauma can impact development in young children. This topic can be understood from various perspectives and domains. In consensus with the living lab discussions, chapter two starts out by elaborating on neurobiological and psychosocial development. Particular attention is paid towards bonding, attachment, mentalization and resilience, as these were identified as meaningful topics by the living lab participants. Bonding and attachment refer to the development of enduring connectedness between an individual and others [12,13]. Mentalization refers to the ability to interpret behavior in relation to internal mental states (e.g., desires, feelings, thoughts, beliefs, intentions or opinions) [28]. Resilience means the ability to bounce back when faced with mishaps and setbacks stemming from a variety of traits (e.g., determination, optimism, faith and hope) [29]. The chapter then goes on to answer the question of how symptoms of toxic stress and trauma might differ among adults and young children. At the end of the chapter the recent vision is underscored by the unique period of opportunity of the first 1000 days when the basis for optimal health, growth and neurodevelopment across the lifespan can be established [30]. 

#### 3.1.3. Chapter 3: How Trauma-Sensitive Care Is Useful When Caring for Young Children

The third chapter focuses on sensitization towards trauma-sensitive care for children aged 0–3 years old. The chapter is written with a focus on trauma-sensitive care on the level of an organization and its policies and structures. It was agreed upon by the living lab members that trauma-sensitive care should be incorporated in the framework of an organization before it can be embraced by the professionals working within the organization. This chapter clarifies how an organization can integrate trauma-sensitive care in its everyday practice (e.g., investing in awareness among all employees). In that sense, trauma-sensitive care should not only be offered to children and to their families, but also to the professional caregivers themselves. Additionally, trauma-sensitive care is explained using six basic principles that allow for customization to different types of cultures and contexts, without forcing prefabricated methods. The principles of trauma-sensitive care are safety, predictability, support, partnership, presence and inclusion. Finally, the chapter highlights how trauma-sensitive care can be implemented structurally (e.g., through systematic screening for trauma; through training and professional development; and through cooperation with other organizations). 

#### 3.1.4. Chapter 4: How to Implement Trauma-Sensitive Care in a Care Setting for Young Children

The fourth chapter of the trauma-sensitive care protocol enhances the understanding of the basic principles by offering reflective tools to childcare professionals, health visitors and pediatricians for both their daily work and moments of crisis. For each of the six basic principles, the chapter invites the reader to reflect on questions from the perspective of the professional caregivers (e.g., “Does the organization actively ensure that childcare professionals feel emotionally safe?” is a reflective question for the safety principle), the perspective of the parent (e.g., “Does the organization pay attention to the family’s own strengths and resources?” is a reflective question for the presence principle), and the child’s perspective (e.g., “Is the daily agenda predictable and familiar to children?” is a reflective question for the predictability principle). Each reflective question is matched with several tips to facilitate the concrete application of the basic principle. The chapter also provides a white space following each principle where notes and reflections can be written down. Additionally, the chapter offers useful practices and methods that can be used to implement trauma-sensitive care in the daily practice and in moments of crisis. Finally, chapter 4 presents handholds for discussing worries with primary caregivers (e.g., parents) following worrying signals, while engaging in trauma-sensitive care. 

#### 3.1.5. Chapter 5: Appendix A

The fifth chapter offers Appendix A to the protocol. A glossary covers an alphabetical list of the key terms that are discussed in the protocol. Additionally, the difference between trauma-awareness, trauma-sensitivity, trauma-responsivity and trauma-”informedness” is explained. In consensus with the experts and living lab participants, the current protocol adopted the term trauma-sensitive care instead of trauma-informed care, because this highlights that the protocol is specifically designed for a general care setting, not aimed at specialized treatment institutions for children coping with trauma. Finally, chapter 5 lists various definitions of trauma from different psychosocial and biological perspectives. 

### 3.2. Training Intervention

As seen in Table 1, 100 childcare professionals took part in the training. Some participants dropped out of the training (8% attrition rate from T0 to T2). Table 2 includes the mean scores and standard deviations for the scales Knowledge, Attitude, Practice (consisting of Infant Care, Parent Interaction, Self-Care and Team environment), Evaluation of the trauma-sensitive care protocol, and General feedback on the training.

Two-way ANOVA (Mm X Country) was used to examine the self-reported differences in scores from Measuring moments (Mm; T0 to T1 to T2) and for all four Countries (Hungary, Belgium, Latvia, and Italy).

First, the effect of self-evaluated Knowledge was assessed for each Measuring moment (Mm) and for every country (Figure 2). The analysis revealed a significant interaction between Mm and Time (F(6, 273) = 3.328, *p* = 0.004). Simple main effects showed a significant effect of Country (F(3, 273) = 37.395, *p* < 0.001) and Mm (F(2, 273) = 113.456, *p* < 0.001) on self-evaluated Knowledge. For Latvia, the increase stagnated between T1 and T2. The biggest difference in self-reported Knowledge was in Italian childcare professionals, especially between T0 and T1. Belgian and Hungarian participants showed a more linear increase.

Second, considering Attitude (Figure 3), the analysis revealed a statistically significant interaction between the effects of Country and Measure moment (F(6, 273) = 3.218, *p* = 0.005). Simple main effects showed that Country had a significant effect on self-reported Attitude (F(3, 273) = 29.318, *p* < 0.001), as did Measure moment (F(2, 273) = 28.904, *p* < 0.001). Further analyses indicated that, even though there was a significant increase in self-reported Attitude, the difference was rather small. Again, the biggest difference over time was in Italian participants. The smallest difference over time was in Hungarian childcare professionals. As the maximum self-reported value was “5”, observations of the mean at T0 indicate that the self-reported Attitude was already high at T0 for Hungary. For Belgian, Italian, and Latvian participants, there was an increase in Attitude from T0 to T1, followed by a plateau.

Third, the effect of self-evaluated Practice was assessed for each Measuring moment (Mm) and for every country (Figure 4). The analysis revealed that there was no statistically significant interaction between the effects of Country and Measure moment (F(6, 273) = 1.111, *p* = 0.356). Simple main effects showed that Country had a significant effect on self-reported Practice (F(3, 273) = 19.849, *p* < 0.001), as did Measure moment (F(2, 273) = 42.095, *p* < 0.001). Childcare professionals from all countries increased significantly in self-reported Practice over time. Only the overall difference of Latvian childcare professionals between T1 and T2 stagnated.

Fourth, considering the Evaluation of the trauma-sensitive care (TSC) protocol (Figure 5), a two-way ANOVA was executed to analyze the effect of Country and Measure moment on the experience with the TSC protocol. The analysis revealed a statistically significant interaction between the effects of Country and Measure moment (F(3, 175) = 2.936, *p* = 0.035). Simple main effects showed that Country had a significant effect on the Evaluation of the TSC protocol (F(3, 175) = 7.870, *p* < 0.001), as did Measure moment (F(1, 175) = 8.597, *p* = 0.004). Further analyses indicated that Belgian and Hungarian childcare professionals expressed an increased positive Evaluation of the TSC protocol between T1 and T2. The experience of the Hungarian childcare professionals was most positive over measuring moments and over Countries. Belgian childcare professionals had the least positive experience, but also expressed an increase. Italian and Latvian childcare professionals similarly expressed a stagnated positive Evaluation of the TSC protocol.

Finally, considering the General feedback on the training (Figure 6), the analysis revealed a statistically significant interaction between the effects of Country and Measure moment (F(3, 175) = 3.343, *p* = 0.021). Simple main effects showed that Country had a significant effect on the General feedback on the training (F(3, 175) = 7.694, *p* < 0.001), as did Measure moment (F(1, 175) = 4.334, *p* = 0.039). Belgian and Hungarian participants showed increased positive General feedback on the training between T1 and T2. For Latvian participants, the evaluation remained stable from T1 to T2. For Italy, the positive General feedback on the training decreased over time.

## 4. Discussion

The first objective of the present study was to develop a trauma-sensitive care protocol for professional caregivers working with children aged 0–3. After exploring the needs and opportunities of the target groups (i.e., professional caregivers), the creation process was oriented towards a combination of knowledge, skills and attitude regarding trauma-sensitive care. Obtaining qualitative information offers unique insight with regard to the professional caregivers’ needs and challenges, thoughts and attitudes. Drawing upon prior literature, the finalized protocol included both theory as well as accessible material aimed at facilitating the translation from theory to practice. Instead of a single all-encompassing source of information, the option was chosen to include a wide variety of reflective exercises, group discussion materials, inspirational practices and a checklist for professional caregivers to use in daily practice. 

As trauma-sensitive care is not a “one size fits all” method, it was important to diversify and to integrate different realities of professional caregivers, from working independently, to being part of a large team. This is in line with the three preconditions to a trauma-sensitive care approach according to Huang et al. [19], that emphasize the need to encourage cultural change on all levels. The first precondition is a concise understanding of the widespread impact of trauma and potential paths to recovery, which is integrated in the protocol through theoretical explanations and reflection exercises on chronic stress, trauma, and their impact, and on how trauma-sensitive care can offer support. The second precondition includes knowledge of the signs and symptoms of trauma from a systemic perspective, which is translated into theoretical exercises based on the professionals’ experiences. A third and final precondition covers an embeddedness of a perspective on health and wellbeing in the policy and general structure of an organization that integrates warmth, support, safety and predictability to prevent and intervene in cases of trauma [31]. This is incorporated through the reflective questions and checklist that make macro-level changes possible. The protocol emphasizes the importance of training amongst professional caregivers and how to approach children aged 0–3 in positive relations [19,20,21]. 

The second objective of the present study was to evaluate the usability and eligibility of the developed protocol. To this end, a quantitative questionnaire was included as an evaluation in a training intervention. The results indicated that the experiences with the protocol were overall positive and remained positive over time. After following the training intervention, the participants self-reported an increase in knowledge, skills and attitude towards trauma-sensitive care for children aged 0–3 years old. Additionally, the professional caregivers evaluated the trauma-sensitive care protocol positively immediately after the training but also at T2, i.e., after three months, suggesting that the trauma-sensitive care protocol evoked a durable positive impact on the participants.

The data of the present study were collected in four different European countries that differ in terms of culture and policy. This implies that similarities found between the countries can potentially be extrapolated to the rest of the European continent. Nevertheless, organizations interested in using the protocol should look into the applicability of the protocol to the local context. If attendance within professional care settings is not the general standard, preventing and addressing the burden of chronic stress by committing to positive relationships in a childcare organization might not be the way to provide trauma-sensitive care in this specific situation. 

While the present study yielded rich data regarding a newly developed trauma-sensitive care protocol, it has several limitations. First, the protocol was assessed in settings where childcare professionals are in regular contact with children aged 0–3 and their parents. The diversification of the sample in terms of type of care setting, frequency of engagement with children and parents, and regularity of the contact between professional and child, will be required to validate the trauma-sensitive care protocol for all professional caregivers working with children aged 0–3. Second, the training and protocol are evaluated only by professionals and only on three measure moments that are close together (i.e., there are three months between T0 and T3). Future studies might benefit from also evaluating the effect of the protocol in a parent sample and by including a more longitudinal measurement (for instance, a year after the training intervention) to assess whether a lasting culture shift towards trauma-sensitive care is achieved. Research agrees that a lack of continuity can undermine the effects of a trauma-sensitive care approach and thus long-term follow-up of compliance with the protocol is necessary [32]. Further, other evaluation methods might also be useful to assess the usability and eligibility of the developed protocol, for instance through observation. These limitations notwithstanding, the present study showed that the developed trauma-sensitive care protocol is a promising tool to introduce trauma-sensitive care principles to professional caregivers working with children aged 0–3 and their families. Strengths of the study include that, to our knowledge, the present study is the first to introduce a trauma-sensitive care protocol aimed at enhancing the knowledge, skills and attitudes of professional caregivers working with children aged 0–3 and their families regarding trauma-sensitive care as a way of supporting those who have experienced chronic stress. The developed protocol and training are open-access, making the materials low-threshold and cost-efficient to professional caregivers who generally lack training and financial support to invest in trauma-sensitive care [16]. As it has been developed and assessed across different countries, we can carefully assume that the trauma-sensitive protocol is applicable in different sectors and different local realities. Moreover, as children who experience adverse experiences can be impacted long-term, for instance through adjustment problems [7,29,33], the protocol and training indirectly contribute to the early identification and prevention of trauma and its consequences. 

In conclusion, professional caregivers working with children aged 0–3 and stakeholders were invited to participate in a living lab in order to develop a trauma-sensitive care protocol for professional caregivers through exploration, co-creation and piloting. The finalized trauma-sensitive care protocol was integrated in a training intervention aimed at professional caregivers. Results are illustrative of the potential of trauma-sensitive care as a method for professional caregivers working with children aged 0–3 and the broader field, such as hospital settings, where chronic stress and trauma might also be addressed, but also for perinatal professional caregivers, who are encouraged to pay attention towards stress and trauma from pregnancy.

## Figures and Tables

**Figure 1 children-10-01035-f001:**
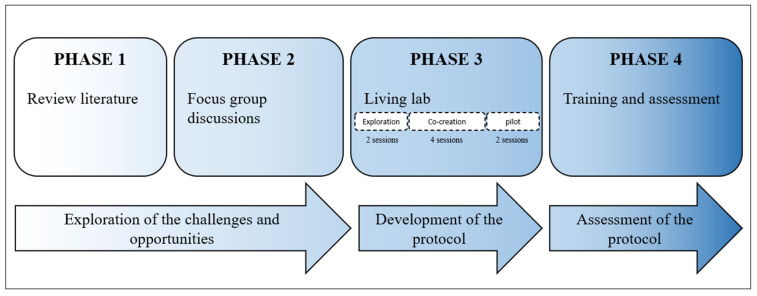
ECLIPS project design. Phase 1 and Phase 2 are integrated in [16].

**Figure 2 children-10-01035-f002:**
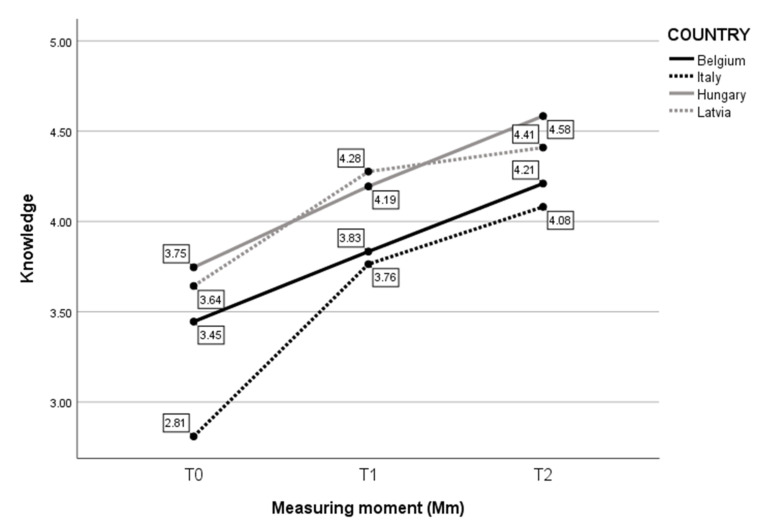
Mean score on self-evaluated Knowledge per Mm and per Country.

**Figure 3 children-10-01035-f003:**
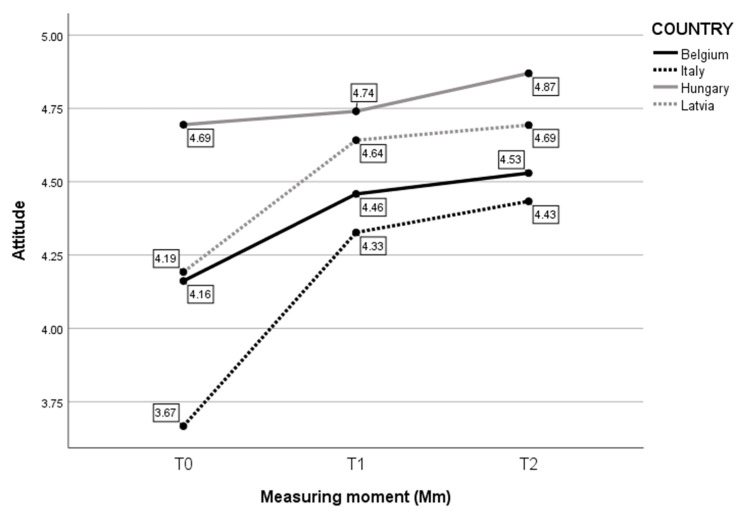
Mean score on self-evaluated Attitude per Mm and per Country.

**Figure 4 children-10-01035-f004:**
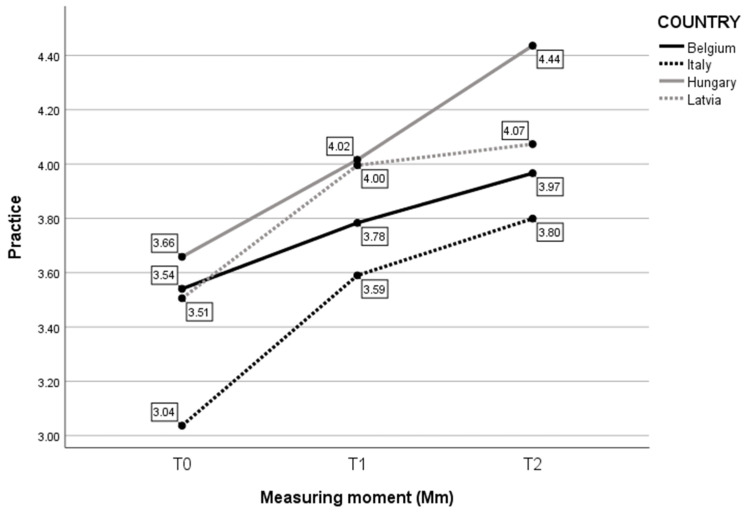
Mean score on self-evaluated Practice per Mm and per Country.

**Figure 5 children-10-01035-f005:**
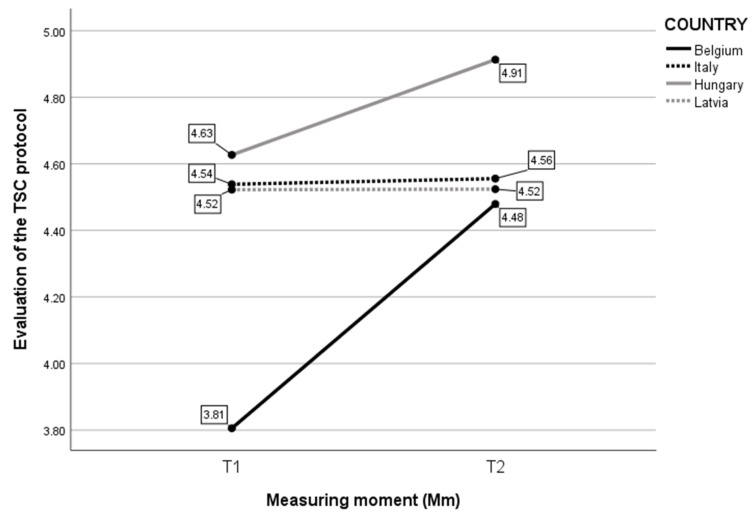
Mean score on self-evaluated experience with the trauma-sensitive care (TSC) protocol per Mm and per Country.

**Figure 6 children-10-01035-f006:**
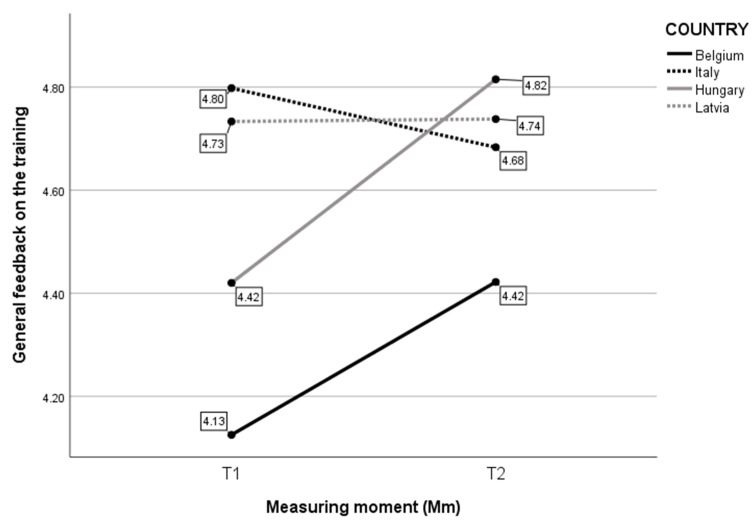
Mean score on self-evaluated General feedback on the training per Mm and per Country.

**Table 1 children-10-01035-t001:** Overview of number of participants in the training intervention.

Project Partner’s Country	T0	T1	T2
Belgium	17	12	17
Italy	30	26	30
Hungary	27	25	23
Latvia	26	30	22
Total sample	100	93	92

Note. All participants were professional caregivers from the childcare sector. T0 = baseline; T1 = one month after the training intervention; T2 = three months after the training intervention.

**Table 2 children-10-01035-t002:** Descriptive data of the T0, T1 and T2 measure moments for all participants (*n* = 100).

	T0		T1		T2		Total			
	M	SD	M	SD	M	SD	M	SD	n items	alpha
Knowledge	3.39	0.62	4.05	0.47	4.31	0.37	3.90	0.64	7	0.797
Attitude	4.17	0.60	4.56	0.46	4.62	0.41	4.44	0.54	4	0.592
Practice	3.41	0.57	3.86	0.94	4.05	0.50	3.77	0.58	13	0.915
Infant care	3.05	0.75	3.65	0.67	3.94	0.61	3.54	0.78	4	0.887
Parent interaction	2.84	0.84	3.54	0.73	3.82	0.66	3.38	0.86	5	0.914
Self-care	3.58	0.65	3.90	0.69	4.16	0.61	3.87	0.69	2	0.460
Team environment	4.16	0.70	4.34	0.63	4.30	0.72	4.27	0.69	2	0.701
Evaluation of the TSC protocol			4.46	0.66	4.63	0.47	4.54	0.58	3	0.885
General feedback on the training			4.59	0.51	4.68	0.46	4.64	0.49	4	0.869

Note. The subscales “Evaluation of the TSC protocol” and “General feedback on the training” were not assessed at T0. TSC = trauma-sensitive care; T0 = baseline; T1 = one month after the training intervention; T2 = three months after the training intervention. M = mean score; SD = standard deviation; n items = number of items; alpha = Cronbach’s alpha.

## Data Availability

The data used and/or analyzed during the current study are available from the corresponding author on request.

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
