# Peer review of "Care When It Counts: Establishing Trauma-Sensitive Care as a Preventative Approach for 0–3-Year-Old Children Suffering from Trauma and Chronic Stress"

_children, 2023, doi:10.3390/children10061035_

Round 1
Reviewer 1 Report
the author wanted to report on their research about Care when it counts: Establishing trauma-sensitive care as a preventative approach for 0–3-year-old children suffering from trauma and chronic stress
This research is good, in presentation manuscript, what do you need to add to your take home massage in this research?
Give us an overview of the advantages and strengths of your research
Author Response
Wednesday 24 May 2023
Manuscript ID children-2348372
Dear reviewer,
Thank you for reviewing our manuscript titled ‘Care when it counts: Establishing trauma-sensitive care as a preventative approach for 0–3-year-old children suffering from trauma and chronic stress’. We are glad to be given the opportunity to revise our manuscript according to the expert suggestions. We hope that our adaptations will cover the feedback from the reviewers. In attachment, you will find the details of our revisions to the manuscript and our responses to the comments. In the manuscript, changes are marked up using the Track Changes function.
We hope that the manuscript can now be considered for publication in Children.
Kind regards,
The authors [anonymous for blinded peer review]

Reviewer 2 Report
General
This manuscript describes the development of a protocol for trauma-sensitive care, aimed at professional caregivers, and its evaluation in different European countries. I really enjoyed reading this manuscript, and I think the authors did a great job – I’m looking forward to seeing the full protocol once your article is published. I have some suggestions that might help to further improve the manuscript, especially regarding the Results and Discussion section.
Abstract
· I have nothing to add, but maybe explain what a “living lab” is – a short explanation in brackets would be enough.
Introduction
· L. 58: What do you mean by “brain development is cumulative”?
· On l. 76, “intensity” of what do you refer to?
· L. 132 ff.: The way I understand this sentence is that professional caregivers were involved in the creation of the protocol, is that correct? If so, I would recommend you mention that a bit earlier in the introduction, as involvement of those with direct experience is a major strength of a project.
· I enjoyed reading your introduction, I think it is well written and makes the important connections. I specifically liked the way you describe the caregiver’s role and the risk of children being misinterpreted as “behaving badly”.
Methods
· I might be wrong, but is Living Lab just a different word for patient / stakeholder involvement? Is there a difference?
· What exactly did you explore in sessions 1 and 2? Current needs or questions of participants?
· On l. 224, you refer to a “self-report questionnaire” – could you please add the name and reference of the questionnaire? Was it designed specifically for the purpose of this study?
Results
· L. 259-268: You mention this earlier, this does not need to be repeated in the Results section. Rather I suggest you start with the link to the open-access protocol and mention the languages in which it is available, and then continue with the sentence: “The finalized protocol…”.
· I suggest you add an example or two to each chapter (as you do for chapter 4). Otherwise it can be difficult to imagine how exactly this might look like in practice (I know you link the open-access protocol, but many readers prefer to have all information in one place).
· L. 370 ff: Please move possible explanations for your results into the Discussion section.
· Personally, I think the first part of your Results section (results of the Living Lab) are more informative and interesting than the second part. I would suggest further summarizing the second part – e.g., I don’t need to know the differences between countries for each subscale of the questionnaire – and use more examples from the protocol (e.g., sentences or exercises) in the first part of the Results section. It’s ok to include the quantitative results if you think that they add something important, but to me, the qualitative bit is much richer.
Discussion
· Regarding my aforementioned point: I like how you summarize the quantitative results in the Discussion section on l. 439-447.
· Maybe you can add results from a longitudinal study that looks at adolescent / adult outcomes in children with ACEs? Either here or in the Introduction. I think that would further strengthen your argument of why this early phase of life is particularly important. My colleagues and I published on this topic a few years ago (please do not feel obligated to cite this specific reference, it’s just one example): Koechlin, Donado et al., 2018: “Effects of Childhood Life Events on Adjustment Problems in Adolescence: A Longitudinal Study”.
· One other thought for the Discussion: early childhood is often neglected finance-wise. A lot more money goes into trying to limit the damage once it’s done in later years of childhood and adolescence (at least in Switzerland, where I live and I think it’s similar in other European countries). Maybe this could be an interesting point to discuss: It seems like your protocol is not cost-intensive, as it is open-access. Can interested professionals just use it by themselves or do they need a training? Either way, it seems to be a low-cost solution.
· I think you are a bit too modest in your Discussion: You included several different voices in the development of your protocol and it seems to be useful for a range of different occupational groups. Given the long-term negative consequences of early stress and trauma, I think this has a lot of potential.
Author Response

(The authors gave the same response as above.)

Reviewer 3 Report
Authors present a timely and relevant investigation that will contribute well to the literature. comments are included to optimize readability and interpretability.
1. Authors appear to use the terms "ACEs" and "traumatic event" interchangeably when these are not synonyms. Several ACEs (e.g., abuse) are traumas (by DSM-5 Category A definition) but some are not (e.g., parent separation/divorce) and rather should be classified as "stressors". I strongly encourage the consistent and most accurate use of terminology in these contexts.
2. line 52 - authors note "trauma symptoms". What might these be? This term is vague for the niave reader. Should explain, especially given the varying use of terminologies noted above (i.e., you cannot have a trauma symptom without a trauma).
3. line 59 - what is meant by "learning behavior"?
4. line 61 - what is meant by "tackle of trauma"? that feels colloquial and is not clear.
5. Line 64+ - another new term "child maltreatment" is introduced...See comments above.
6. Line 86 - "chronic stress"...see comments above
7. Unclear of why the open access trauma protocol is blacked out (perhaps the journal did this). Would have been very helpful to see this.
8. Would be interesting and helpful for authors to emphasize the caregivers perspective in trauma-informed care delivery. It can be stressful in and of itself to care for maltreated children (as authors cited in the intro). Did this trauma informed care training positively impact how prepared or efficacious these caregivers felt?
Author Response

(The authors gave the same response as above.)

Round 2
Reviewer 3 Report
Authors have adequately addressed all comments.